

# Plastome of mycoheterotrophic *Burmannia itoana* Mak. (Burmanniaceae) exhibits extensive degradation and distinct rearrangements

Xiaojuan Li[1,2], Xin Qian[1,2], Gang Yao[3], Zhongtao Zhao[1] and Dianxiang Zhang[1]

[1] Key Laboratory of Plant Resources Conservation and Sustainable Utilization, South China Botanical Garden, Chinese Academy of Sciences, Guangzhou, China
[2] College of Life Sciences, University of Chinese Academy of Sciences, Beijing, China
[3] South China Limestone Plants Research Center, College of Forestry and Landscape Architecture, South China Agricultural University, Guangzhou, China

Corresponding authors
Zhongtao Zhao,
zhzht621@scbg.ac.cn
Dianxiang Zhang,
dx-zhang@scbg.ac.cn

## ABSTRACT

Plastomes of heterotrophs went through varying degrees of degradation along with the transition from autotrophic to heterotrophic lifestyle. Here, we identified the plastome of mycoheterotrophic species *Burmannia itoana* and compared it with those of its reported relatives including three autotrophs and one heterotroph (*Thismia tentaculata*) in Dioscoreales. *B. itoana* yields a rampantly degraded plastome reduced in size and gene numbers at the advanced stages of degradation. Its length is 44,463 bp with a quadripartite structure. *B. itoana* plastome contains 33 tentatively functional genes and six tentative pseudogenes, including several unusually retained genes. These unusual retention suggest that the inverted repeats (IRs) regions and possibility of being compensated may prolong retention of genes in plastome at the advanced stage of degradation. Otherwise, six rearrangements including four inversions (Inv1/Inv2/Inv3/Inv4) and two translocations (Trans1/Trans2) were detected in *B. itoana* plastome vs. its autotrophic relative *B. disticha*. We speculate that Inv1 may be mediated by recombination of distinct tRNA genes, while Inv2 is likely consequence of extreme gene losses due to the shift to heterotrophic lifestyle. The other four rearrangements involved in IRs and small single copy region may attribute to multiple waves of IRs and overlapping inversions. Our study fills the gap of knowledge about plastomes of heterotroph in *Burmannia* and provides a new evidence for the convergent degradation patterns of plastomes en route to heterotrophic lifestyle.

## INTRODUCTION

Plastids were derived from a common cyanobacterial ancestor that established a permanent endosymbiotic relationship with mitochondriate ancestor (*Gould, Waller & McFadden, 2008*; *Ku et al., 2015*). Over the evolutionary history, the plastid encoding genes have undergone multiple transfers and losses which are ongoing processes. For instance,

the ancestor genes of the plastid have transferred to other two genomes (nuclear and mitochondria) or total lost during evolution (*Martin & Herrmann, 1998*; *Cusimano & Wicke, 2016*). Most plastomes of land plants are canonical circular and quadripartite including two inverted repeats (IRs) separated by larger single copy region (LSC) and small single copy region (SSC). Most typical plastomes are variable in size ranging from approximate 120 to 170 kbp with about 113 unique genes including about 79 protein-coding genes, four rRNA genes, and 30 tRNA genes (*Wicke et al., 2011*). These genes are classified into three main classes depending on functions: (1) photosynthesis related genes (ndh/atp/psa/psb/pet/ycf3/ycf4/rbcL); (2) transcription, transcript maturation, and translation related genes (rpo/infA, matK, and tRNAs/rRNAs/rps/rpl), (3) other non-bioenergetic function genes (accD/clpP/ycf1/ycf2/ccsA/cemA) (*Bock, 2007*; *Wicke et al., 2011*).

Parasitic and mycoheterotrophic plants, which establish a physiological relationship with either plants or fungi to obtain organic and mineral nutrients, are referred to have partially or fully lost the capacity of photosynthesis (*Wicke & Naumann, 2018*). The transition of lifestyle from autotrophic to heterotrophic may have left measurable clues in genomes, such as adaptive and non-adaptive changes (*Wicke et al., 2016*). Studies about plastomes of heterotrophs revealed that plastomes of heterotrophic plants experienced convergent degradation syndromes compared with those of autotrophs, such as overall decreases in genome size along with functional and physical gene losses, decrease in GC-content, increased frequency of rearrangements, accumulation of indels, and losses of introns (*Wicke et al., 2013*; *Logacheva et al., 2014*; *Lam, Soto Gomez & Graham, 2015*; *Lim et al., 2016*; *Wicke et al., 2016*; *Petersen et al., 2018*; *Schneider et al., 2018*).

Hitherto, several conceptual models of plastome degradation have been proposed to account for the order of the heterotrophs plastomes degradation in a simplified and idealized manner. In general, plastome degradation follows five major stages (*Barrett & Davis, 2012*; *Barrett et al., 2014*; *Naumann et al., 2016*; *Wicke et al., 2016*; *Graham, Lam & Merckx, 2017*): (1) ndh genes will be affected first and functionally lost from the plastomes once plants have the ability of obtaining nutrients using heterotrophic way; (2) when the lifestyle transfer to obligate heterotrophic, most of photosynthesis related genes and some housekeeping genes are lost; (3) atp/rbcL and nonessential housekeeping genes are lost or functionally replaced; (4) other nonbioenergetic genes (e.g., accD, clpP, ycf1/2) are lost or functionally replaced; (5) all plastid encoding genes are lost.

*Burmannia* contains about 60 species spanning from autotrophic, hemi-mycoheterotrophic to mycoheterotrophic (*Jonker, 1938*; *Zhang, 1999*; *Wu, Zhang & Saunders, 2010*), which could provide an excellent model system for understanding the evolution of plastome responding to the lifestyle shift from autotrophic to heterotrophic. Although the plastome of autotrophic *Burmannia disticha* (*Ma, Ma & Li, 2018*) has been described, the plastome of mycoheterotrophic species in *Burmannia* has never been reported, hindering our efforts in elucidation of plastid evolution in the genus. Aiming to provide new evidence in our understanding of the mechanism of plastid evolution in mycoheterotrophic angiosperms, here we focus on *B. itoana*, a mycoheterotrophic perennial herb (Fig. 1) distributed in the coastal provinces of southern China and Ryukyu

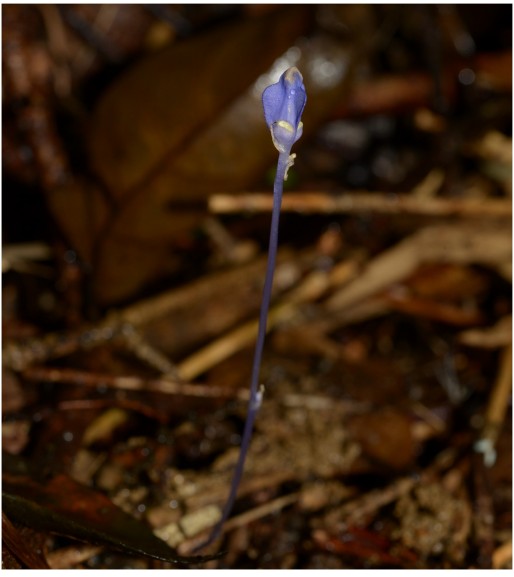

**Figure 1** ***Burmannia itoana* in flower, Long Men Park, Guang Dong, China.**

Islands of Japan (*Jonker, 1938*; *Zhang, 1999*; *Wu, Zhang & Saunders, 2010*). We sequenced the *B. itoana* plastome and examined its content and structure. Furthermore, we compared the plastome of *B. itoana* with those of its four documented relatives (*B. disticha*, *Thismia tentaculata* (Thismiaceae), *Tacca chantrieri* (Taccaceae), and *Dioscorea zingiberensis* (Dioscoreaceae)) in Dioscoreales.

# MATERIALS AND METHODS

## DNA extraction and sequencing

Samples of *B. itoana* were collected from the Longmen Park, Guangdong Province, China. The voucher specimens (LXJLM07) are deposited in the IBSC (Herbarium of South China Botanical Garden). Total DNA was extracted from an individual plant with an identical collection number as that of voucher specimens using a DNeasy Plant Mini Kit (Qiagen, Hilden, Germany). The total DNA was used to generate libraries with average insert size of 500 bp and sequenced using Illumina HiSeq 2000 with 150 bp paired-end read lengths.

## Plastome de novo assembly and annotation

The plastome was assembled using CLC Genomics Workbench v9.0 (CLC BIO, Aarhus, Denmark) with parameters as follows: wordsize 63, bubble size 50, minimal contig length 1,000 bp. Three plastome-like contigs were picked out through mapping all assembled contigs to the sequence of *B. disticha* plastome, and then these contigs were merged using Geneious (version 11.1.5) (*Kearse et al., 2012*) to build draft plastomes. Specific primers were designed to confirm the overlap of the aligned contigs and identify the borders of the LSC, SSC, and IRs regions using through PCR and Sanger sequencing method with these primers as follows. LSC/IRa-F (5′ TGA GAC CTA GTG CTC AAG GGA 3′),

LSC/IRa-R (5′ AAG GTT TAA GAT TTG TAT TTG AAA GA 3′); LSC/IRb-F (5′ CGA GTC ACA CAC TAA GCA TAG C 3′), LSC/IRb-R (5′ TGA GAC CTA GTG CTC AAG GGA 3′); SSC/IRa-F (5′ TCT CTT TAA CAT TTA TGA CAC GAC A 3′), SSC/IRa-R (5′ TTG CGA ACA TAC TCC CCA GG 3′); SSC/IRb-F (5′ AAC GCG TTA GCT ACA GCA CT 3′), SSC/IRb-R (5′ TGA GTT AGT GTG AGC TTA TCC 3′). Validated complete plastome was annotated using GeSeq (*Tillich et al., 2017*) with default sets. Furthermore, tRNAs were predicted using tRNAscan-SE (*Schattner, Brooks & Lowe, 2005*). Finally, the plastome map was visualized using OGDRAW v. 1.2 (*Lohse, Drechsel & Bock, 2007*). The final annotated plastome was deposited in GenBank under accession number MK318822, and annotation of plastome is summarized in Additional File S1.

## Data collection and comparative analysis of plastomes

Four reported plastomes of *B. itoana* relatives including *B. disticha* (Burmanniaceae), *Thismia* tentaculata (Thismiaceae), *Tacca chantrieri* (Taccaceae), and *D. zingiberensis* (Dioscoreaceae) were downloaded from GenBank, and the Genbank accession is MG792012, KX171421, KX171420, and NC_027090, respectively.

The GC-content of plastome and the four junctions of LSC/IRB, LSC/IRA, SSC/IRB, and SSC/IRA were identified using Geneious version 11.1.5 (*Kearse et al., 2012*). Gene contents of plastomes were compared between *B. itoana* and four reported species in Dioscoreales. Gene orders excluding copies in IRs were explored using Mauve (*Darling et al., 2004*) plugged in Geneious.

## RESULTS

### General characteristics of *B. itoana* plastome

*Burmannia* itoana plastome represents a quadripartite circular molecule containing two larger IR regions (IRA/IRB: 12,174 bp) separated by a large single-copy region (LSC: 18,441 bp) and a small single-copy region (SSC: 1,674 bp) (Fig. 2). The *B. itoana* plastome is 44,636 bp in length with 39 genes (Figs. 2 and 3) including 33 putatively functional genes and six putative pseudogenes. The presumably functional genes include four rRNA genes, eight tRNA genes and 21 protein genes, while six genes (petG, rpl36, trnH_GUG, trnD_GUC, trnG_GCC, and trmS_UGA) are identified as putative pseudogenes based on the reasons below: the presence of internal stop codons in petG and rpl36; the anticodon sequences of the trnH_GUG was deleted compared with the typical trnH_GUG; multiple base changes in anticodon and other regions in trnD_GUC compared with the functional trnD_GUC, and no tRNA was predicted by tRNAscan; several indels or substitutes in the trnG_GCC compared with the typical trnG_GCC, and no tRNA was predicted by tRNAscan; several indels or substitutes in the anticodon and other regions of the trmS_UGA compared with the typical trmS_UGA, and no tRNA was predicted by tRNAscan. The rpl16, rpl2, and rps12 each contains one intron (the 3-end intron of rps12 is absent), while the clpP gene harbors two introns. The overall GC-content of *B. itoana* plastome is 32%, which is lower than those of its autotrophic relatives but is higher than that of *Thismia tentaculata*. The GC-content of IR, LSC, and SSC in *B. itoana* is 37.8%, 24.5%, and 32.4%, respectively (Table 1).

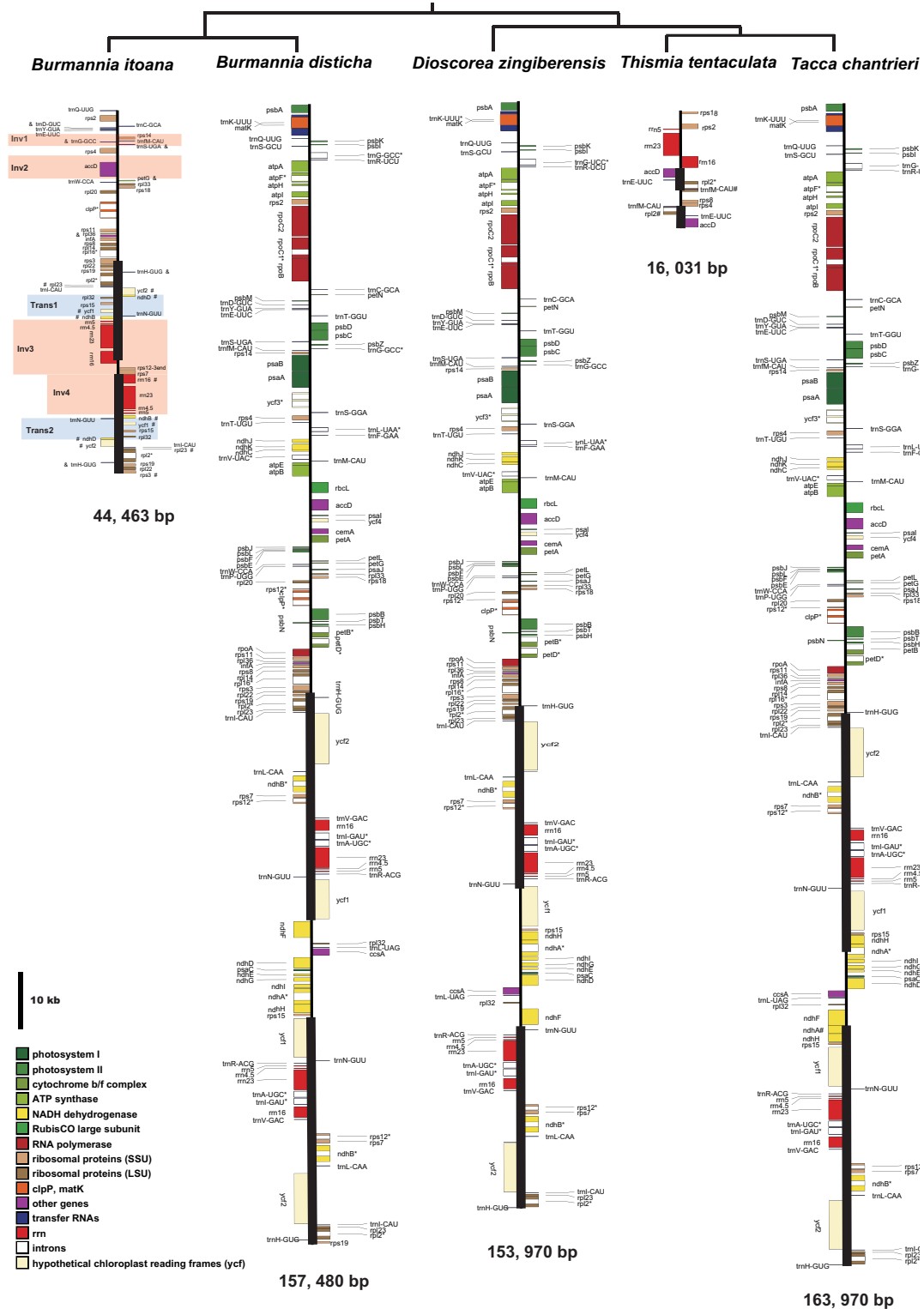

**Figure 2 Plastome structure of five species in Dioscoreales.** All genes are colored according to functional complexes. Genes shown left the line are transcribed counterclockwise, those right the line are transcribed clockwise. The light red blocks show inversions and light blue blocks show insertions. Genes containing intron(s) are marked with the symbol *. Pseudogenes and truncated genes are marked with "&" and "#," respectively. The inverted repeats (IRs) are shown with bold lines.

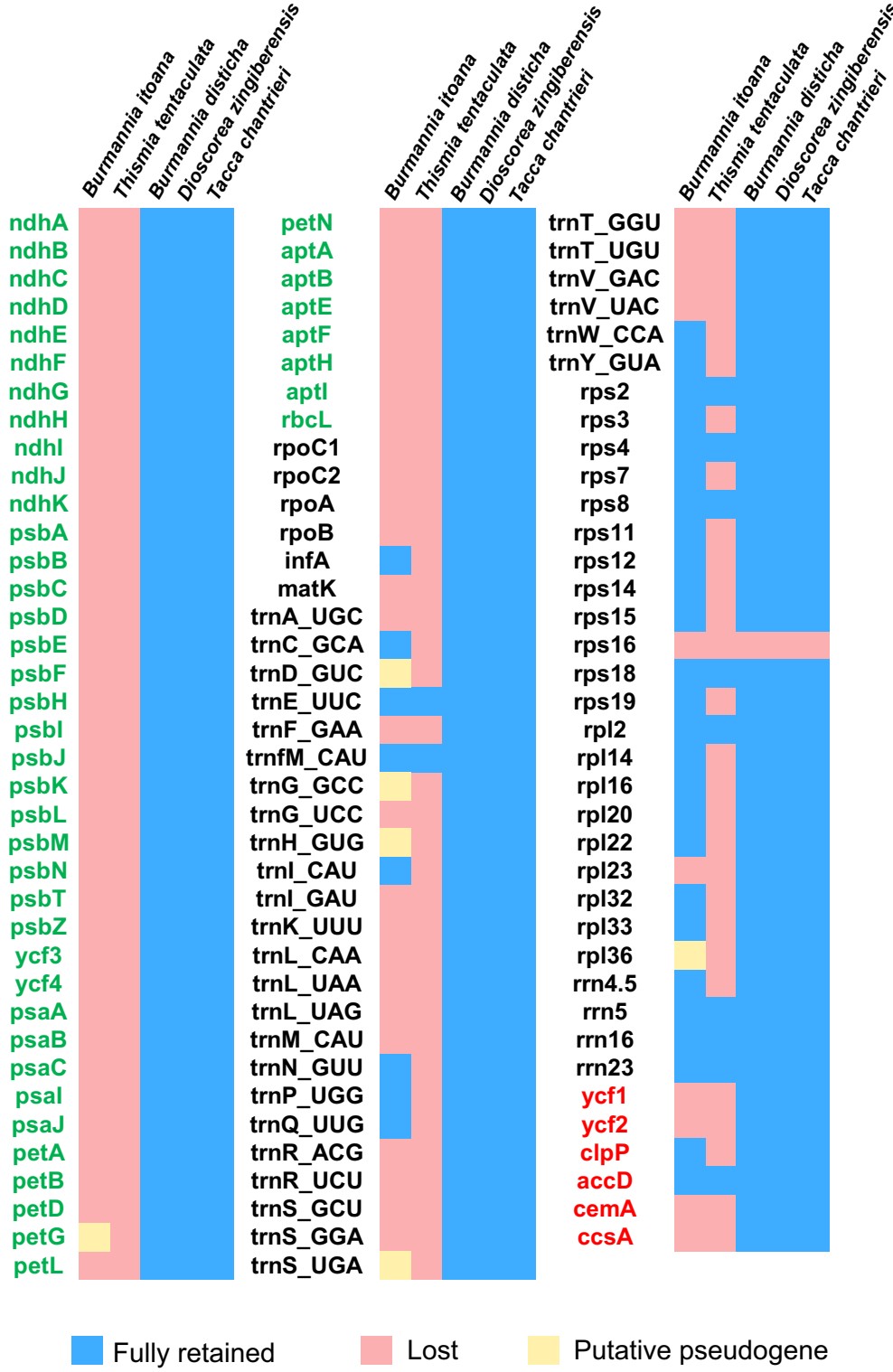

**Figure 3** **Heat map showing gene contents in five species in Dioscoreales.** Genes in blue are retained and presumed fully functional; those in red are absent and those in yellow are putative pseudogenes. Gene names of photosynthesis related genes, genetic apparatus genes, and other genes are in green, black, and red, respectively.

**Table 1 General information of plastomes from five species in Dioscoreales.**

| Taxon | Length (bp)/GC-content (%) | | | | Gene number | | |
|---|---|---|---|---|---|---|---|
| | Total | IR | LSC | SSC | rRNAs | tRNAs | Protein coding |
| *Burmannia itoana* | 44,463/32.0 | 12,174/37.8 | 18,441/24.5 | 1,674/32.3 | 4 | 12 | 23 |
| *Burmannia disticha* | 157,480/34.9 | 31,616/39.5 | 81,231/32.3 | 13,017/28.8 | 4 | 30 | 78 |
| *Dioscorea zingiberensis* | 153,970/37.2 | 25,491/43.0 | 83,950/35.1 | 19,038/31.2 | 4 | 30 | 78 |
| *Tacca chantrieri* | 163,007/36.7 | 33,837/40.3 | 85,241/34.7 | 10,092/30.6 | 4 | 30 | 78 |
| *Thismia tentaculata* | 16,031/27.2 | 2,948/30.0 | 7,799/29.1 | 2,336/13.7 | 3 | 2 | 7 |

## Comparative analyses of gene contents and gene order

Each plastome of the autotrophic relatives *B. disticha*, *Tacca chantrieri*, and *D. zingiberensis* contains 112 genes (Fig. 3). The second intron of rps12 is lost in *B. itoana* plastome, whereas rps12 is totally absent from the *Thismia* tentaculata plastome, while this gene is retained in each plastome of the autotrophic relatives. Like plastome of *Thismia* tentaculata, *B. itoana* plastome exhibits not only extensive reduction in length but also losses of most photosynthesis related genes and some housekeeping genes, while only one photosynthesis related gene, named petG, is retained as a pseudogene in *B. itoana* plastome. In addition, *B. itoana* plastome shares all of the 12 genes retained in *Thismia* tentaculata plastome. Compared with *B. itoana* plastome, *Thismia* tentaculata plastome has experienced further degradation.

Alignment using Mauve showed that plastomes of autotrophs are highly colinear with exceptions that the inversion of the SSC in *B. disticha*, and extremely modified plastome of *Thismia* tentaculata (Fig. 4). Here, we mainly focus on the comparison between the plastome structure of *B. itoana* and *B. disticha* represented in Figs. 2 and 4. The results showed that gene order of *B. itoana* plastome exhibits six rearrangements including four inversions (Inv1, Inv2, Inv3, Inv4) and two translocations (Trans1/Trans2) vs. that of *B. disticha* plastome. Inv1 contains the cluster of trnS_GUA—trnG_GCC—trnfM_CAU—rps14 (ca. 1,282 bp), and Inv2 just contains accD (ca. 2,725 bp), both of which are located in LSC. Inv3 contains the cluster of rps7—rps12—rrn16—rrn23—rrn4.5—rrn5 (ca. 6,678 bp) stretching across the IRs and SSC, and Inv4 contains the cluster of rrn16—rrn23—rrn4.5—rrn5 (ca. 5,007 bp) located in IRs. Trans1/Trans2 contain the cluster of partial truncate ndhD—rpl32—rps15—truncate ycf1—trnN (ca. 2,600 bp), andTrans1/Trans2 are involved in the IRs and SSC of *B. disticha* plastome.

## Boundaries of SC/IRs of Dioscoreales plastomes

The boundary of LSC/IRB in *B. itoana* plastome slightly expands to rps3, which is different from that in its autotrophic relatives. Specifically, boundary of LSC/IRB in *B. disticha* is located in rpl22, and those of *Tacca* chantrieri and *D. zingiberensis* are located in the intergenic space between rps19 and trnH_GUG. The boundary of SSC/IR in *B. itoana* plastome resides in rps7. In contrasting to that, the boundaries of SSC/IR are located in the intergenic of rps15 and ycf1, ycf1 in *B. disticha* and *D. zingiberensis*, respectively. In *Tacca chantrieri*, one boundary of SSC/IRs expands to the intron of ndhA. In summary, IRs have

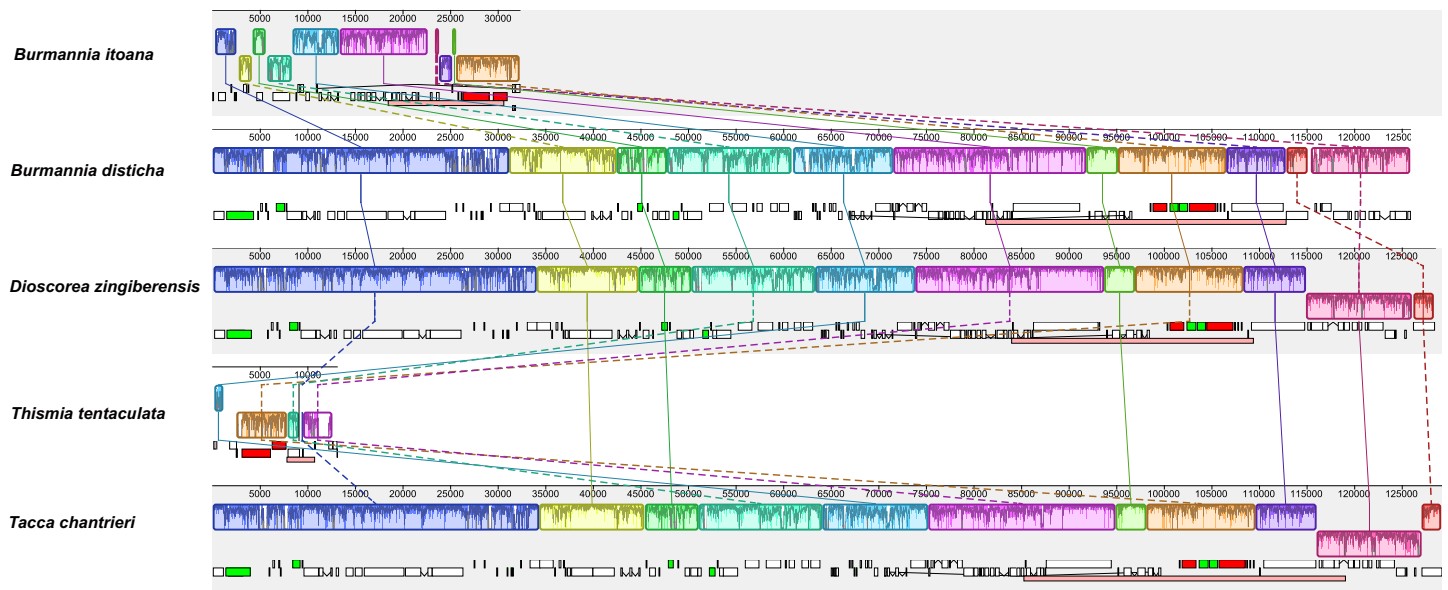

**Figure 4 Synteny of plastomes from five species.** The dashed lines and solid lines illustrate rearranged gene blocks and collinear between the two plastomes, respectively.

undergone expansions in both plastomes of *B. itoana* and *B. disticha* relative to those in other compared lineages. The boundaries of SC/IRs of Dioscoreales plastomes are showed in Fig. 2.

## DISCUSSION

### Convergent evolution of plastomes in heterotrophs

Accompanying with forfeiting of photosynthetic capabilities, the plastomes of heterotrophic plants undergo varying degrees of degradation in size and gene contents (*Wicke et al., 2013*, *2016*). For instance, the largest plastome is that of heterotrophic chlamydomonadalean alga *Polytoma uvella* with about 230 kb in length (*Figueroa-Martinez et al., 2017*), while the smallest sequenced plastome is 11,348 bp from endoparastic *Pilostyles aethiopica* (*Bellot & Renner, 2015*). Moreover, previous studies reported that plastome has possibly been totally lost in green algal genus *Polytomella* (*Smith & Lee, 2014*) and land flowering plant *Rafflesia lagascae* (*Molina et al., 2014*). Our results show that *B. itoana* plastome displays a typical quadripartite architecture with LSC and SSC separated by IRs consistent with that of canonical land plant plastomes. The length of *B. itoana* platome is only one-third of its autotrophic relative *B. disticha* and is nearly three times that of heterotrophic *Thismia* tentaculata. As previously mentioned, gene losses in plastomes of heterotrophs follow five stages (*Barrett & Davis, 2012*; *Barrett et al., 2014*; *Naumann et al., 2016*; *Graham, Lam & Merckx, 2017*; *Wicke & Naumann, 2018*). Accordingly, *B. itoana* plastome is likely at the fourth stage with non-bioenergetic and housekeeping genes (e.g., ycf1/2, rps/rpl/tRNAs) being physically or functionally lost. Losses of ycf1 and ycf2 in *B. itoana* plastome support that a general core set of genes (ycf1, ycf2, accD, clpP, infA, and trnE) varies among heterotrophs (*Bellot & Renner, 2015*; *Schelkunov et al., 2015*; *Lim et al., 2016*; *Naumann et al., 2016*; *Roquet et al., 2016*).

It is a well-accepted tenet that matK encodes one kind of protein required for group IIA intron splicing of seven plastid genes trnV_UAC, trnI_GAU, trnK_UUU, trnA_UGC, rpl2, rps12 (3-end intron), and atpF (*Liere & Link, 1995*; *Zoschke et al., 2010*). Normally, the group IIA intron will not be spliced out from the transcripts without matK. Parallel losses of matK and group IIA intron-containing genes have been reported in *Cuscuta* (*McNeal et al., 2009*). In *B. itoana* plastome, matK is physically lost accompanying with the loss of other six genes containg group IIA intron; however, rpl2 and rps12 (the second intron of rps12 is lost) are retained. Similarly, matK is absent or truncated in the plastomes of heterotrophic *Cynomorium coccineum*, *Rhizanthella gardneri*, and *Epipogium aphyllum*, while the second intron of rps12 is lost but rpl2 remains intact (*Delannoy et al., 2011*; *Bellot & Renner, 2015*; *Schelkunov et al., 2015*). However, the introns of both rpl2 and rps12 are still sustained while matK are absent in *Hydnora visseri* plastome, and all rps12 and rpl2 are transcribed properly (*Naumann et al., 2016*). In addition, a large proportion of matK in some species of orchids are pseudogenes while all of the seven genes with group II introns are retained (*Kim, Kim & Kim, 2014*; *Feng et al., 2016*). These instances of matK loss or pseudogenization with retention of group IIA introns may be because that the function of matK could be recovered by that of genes from nucleus (*Mohr & Lambowitz, 2003*), or some group IIA introns may be able to self-splice (*Kim, Kim & Kim, 2014*).

Most reported plastomes at the advanced stage of degradation have experienced multiple losses of tRNA genes reviewed in *Graham, Lam & Merckx (2017)* and *Wicke & Naumann (2018)*, while plastomes of autotrophs typically harbor 30 tRNAs. Combinations of reported plastomes exhibit that the number of tRNAs retained in plastomes at the advanced stage of degradation ranges from zero in two endoparasitic species of *Pilostyles* (*Bellot & Renner, 2015*) to 24 in *Orobanche gracilis* (Genbank accession number: NC_023464, Orobanchaceae). Without exception, only 12 tRNAs (eight functional and four pseudogenized) are retained in the *B. itoana* plastome. In addition, comparative analyses show that most plastomes contain a unique set of tRNAs in heterotrophs. For instance, the subset of tRNAs remained in *B. itoana* plastome has not been detected in others. It shares only eight tRNAs with *Rhizanthella gardneri* (containing 10 tRNAs) (*Delannoy et al., 2011*), seven with *Epipogium* roseum (containing eight tRNAs) (*Schelkunov et al., 2015*), and six with *Sciaphila densiflora* (containing six tRNAs) (*Lam, Soto Gomez & Graham, 2015*). Moreover, a question is raised that tRNAs remained in plastomes of heterotrophs are due to adaptive selection or have just fortuitously escaped deletion. Computer simulations suggest that about half of tRNAs in plastome of *Epifagus virginiana* are retained by chance and half are maintained by selection (*Lohan & Wolfe, 1998*). It was a plausible explanation that the two tRNAs (trnE_UUC and trnfM_CAU) are essential for heme biosynthesis and mitochondrial protein synthesis, respectively, and cannot be replaced with their cytosolic orthologues (*Barbrook, Howe & Purton, 2006*). Previous study showed that two tRNAs (trnE_UUC and trnfM_CAU) are shared within most heterotrophs. Without exception, *B. itoana* plastome shares two tRNAs (trnE_UUC and trnfM_CAU) with those of *Thismia* tentaculata (*Merckx et al., 2017*) and *Sciaphila thaidanica* (*Petersen et al., 2018*), both of which only harbor the two tRNAs. Cryptically,
it could not explain why no functional tRNAs was detected in two species of *Pilostyles* (*Bellot & Renner, 2015*). Therefore, the evolution of the tRNAs in plastomes is cryptic and remains overlooked. Taken together, loss of tRNAs is likely species-/lineage-specific and undergoes varying degrees within heterotrophs.

## Unusual retention of genes in *B. itoana* plastome

Generally, plastomes of heterotrophs which at the fourth stage of degradation begin to shed all coding regions for photosynthetic pathway (*Barrett et al., 2014*; *Graham, Lam & Merckx, 2017*; *Wicke & Naumann, 2018*). Attractively, petG which encodes a subunit protein of the cytochrome b6/f complex for connection of PSI and PSII (*Bock, 2007*; *Wicke et al., 2011*), is idiosyncratically retained as a pseudogene in *B. itoana* plastome but is not detected in other heterotrophs at the advanced stage of degradation. One hypothesis suggest that proximity to the essential gene or essential function can help genes escaping from being lost (*Lohan & Wolfe, 1998*). In plastome of *B. itoana*, petG is proximate to trnW_CCA and accD, two genes could prolong retention in plastome of most heterotrophs. However, petG is totally lost while trnW_CCA and accD are intact in plastomes of *Rhizanthella* gardneri and *Sciaphila* densiflora (*Delannoy et al., 2011*; *Lam, Soto Gomez & Graham, 2015*). In addition, *Naumann et al. (2016)* suggested that if plastome harbors pseudogenes, degradation of plastome would be ongoing. Holding six pseudogenes implied that plastome of *B.itoana* may undergo degradation, and these pseudogenes including petG would be totally deleted.

In heterotrophs, the ribosomal genes begin to be absent from plastomes with varying degrees at the advanced stages (*Barrett & Davis, 2012*; *Barrett et al., 2014*; *Naumann et al., 2016*; *Wicke et al., 2016*; *Graham, Lam & Merckx, 2017*). So far, no study reported functional or physical loss of rpl36 while rpl32, rpl20, rpl22, rpl23, and rpl33 are retained. Interestingly, rpl36 become a pseudogene in *B. itoana*. Some factors, such as short length, location in conserved operon, or difficulty to be replaced by plastid compartments, may contribute to the retention of genes (*Lohan & Wolfe, 1998*; *Wicke et al., 2013*, *2016*). Commonly, rpl36 is located in the intergenic space of rps11 and infA in the most conservative operon including almost all of ribosomal protein-coding genes. Therefore, residing in essential operon and adjacent essential genes could not prevent rpl36 from being pseudogenized in *B. itoana* plastome. It is plausible to speculate that the function of rpl36 is not essential, or that its loss can be compensated by other intracellular genomes in *B. itoana* plastome (*Schelkunov et al., 2015*; *Cusimano & Wicke, 2016*; *Naumann et al., 2016*; *Petersen et al., 2018*).

Among the ribosomal genes, rps15 and rpl32 are the first batch of genes to be absent from reported plastomes at the advanced stage of degradation. However, rps15 and rpl32 are still retained and putatively functional in *B. itoana* plastome. Previous study suggested that there is a strong positive correlation between the number of putatively functional genes and plastome length among heterotrophic angiosperms (*Barrett & Kennedy, 2018*). To date, the two genes rps15 and rpl32 are only preserved in reported plastomes whose length are longer than 80 kb only except the *Monotropa hypopitys* plastome with ca. 40 kbp in length (*Ravin et al., 2016*). In *B. itoana* plastome, rps15 and rpl32 are resided in IRs,

typically located in SSC. Previous studies proved that two copies of genes in IR offer more opportunities to correct the aberrant mutations (*Zhu et al., 2016*; *Choi et al., 2018*). Thus, we could reasonably speculate that IRs may shelter these genes from being deleted in *B. itoana* plastome.

## Multiple origination of rearrangements in *B. itoana* plastome

It has been showed that some rearrangements are correlated with intermolecular recombination between distinct tRNA (*Hiratsuka et al., 1989*; *Haberle et al., 2008*; *Barrett & Kennedy, 2018*). In plastomes of autotroph *B. disticha*, the cluster in Inv1 is part of the cluster of trnS_UGA—psbZ—trnG_GCC—trnfM_CAU—rps14—psaB—psaA—ycf3—trnS_GGA. Thus, we postulate that the recombination of trnS_UGA and trnS_GGA may result in the Inv1 of *B. itoana* plastome as the 39-kb inversion meditated by a pair of 29-bp IRs located in the trnS_GGA and trnS_GCU in *Tylosema fassoglensis* plastome (*Wang et al., 2018*). Previous comparative analysis between plastome of heterotrophic plant *Petrosavia stellaris* and that of its photosynthetic relative *Japonolirion osense* suggested that the rearrangements in the *Petrosavia* stellaris plastome are likely associated with transition to heterotrophic way of life (*Logacheva et al., 2014*). Herein, the accD (Inv2) in the plastome of *B. itoana* is inverted vs. that in its photosynthetic relative *B. disticha*. Inversion of accD is not only absent from the plastomes of heterotrophic plants *Petrosavia* stellaris and *Sciaphila* densiflora (*Logacheva et al., 2014*; *Lam, Soto Gomez & Graham, 2015*), but also has occurred independently in many autotrophic plants, e.g., *Tylosema esculentum* (Fabaceae, KX792933), *Passiflora edulis* (Passifloraceae, KX290855), and *Scaevola taccada* (Goodeniaceae, MK397896). In autotrophic plants, the inversion of accD is mostly resulted from the inversion of a block containing accD and other genes. Commonly, accD is located in a block mainly consisting of the photosynthesis related genes, most of which were lost in *B. itoana* plastome. It needs more studies to hypothesize if gene losses have triggered the inversion along with the shift from autotroph to heterotroph.

In addition, few rearrangements are located in the middle of IRs in plastomes of land plants restricted rare lineages, such as some ferns (*Robison et al., 2018*), lycophyte (*Mower et al., 2019*), Geraniaceae (*Weng et al., 2014*). Interestingly, four rearrangements consisting of two inversions (Inv3/Inv4) and two translocations (Trans1/Trans2) involved in the IRs and SSC are identified in *B. itoana* plastome. The two inversions, which are typically located in IRs, are detected across the IRs and SSC. The two translocations (Trans1/Trans2), which are typically located in the junction of IRs and SSC, are translocated into IRs of *B. itoana* plastome. It was hypothesized that Mobile Open Reading Frames in Fern Organelles elements are regularly associated with inversions and changes to the IRs (*Robison et al., 2018*). In our study, the patterns of these rearrangements are analogous to those of rearrangements in several plastomes of ferns (*Robison et al., 2018*). However, there is no unusual open reading frames (ORF) in the *B. itoana* plastome, while unusual ORF was identified in the ferns plastomes. Hence, we suggest that the four rearrangements in *B. itoana* plastome could not be consequence of mobile element as in the ferns plastome. Likewise, gene relocation was detected in the plastomes of Oleaceae

(*Jasminum* and *Menodora*) (*Lee et al., 2007*) and lycophyte (*Mower et al., 2019*). Both studies suggested that gene relocation attributes to overlapping inversions rather than the direct transposition or intragenomic translocation. Therefore, we speculate that the four rearrangements detected in IRs and SSC of *B. itoana* plastome may be resulted from multiple expansions and contractions of IRs as in the plastome of *Pelargonium hortorum* (*Chumley et al., 2006*) and overlapping inversions as those in *Jasminum*, *Menodora*, and lycophyte. Firstly, several blocks in SSC (e.g., rpl32, rps15, and ndhD) translocated into the IRs resulting from expansion of IRs in the plastome from the ancestor of *B. itoana*. Then the block including these genes in the four rearrangements of *B. itoana* plastome was inverted, which could result in the translocations (Tans1/Trans2) including rps15, rpl32, ndhD, trnN, and ycf1. After that, one inversion may occur between the first exon of ndhB and Tans1/Trans2, and then the first exon of ndhB could be inverted once again. Otherwise, contraction of IRs could explain the translocation and loss of one copy of the block including rps7 and rps12 in Inv3. If intermediate steps are detected in the plastomes of *Burmannia* lineages as those in lycophyte (*Mower et al., 2019*), then these could provide a better explanation for that the multiple waves of the IRs and overlapping inversions contribute to the four rearrangements in *B. itoana* plastome.

## CONCLUSION

Consistent with the convergent evolution of plants plastomes en route to heterotroph, *B. itoana* plastome exhibits rampant degradation and distinct rearrangements. Based on models of plastome evolution, *B. itoana* plastome is at the advanced stages of degradation, and occurrence of six putative pseudogenes suggests that it could undergo further degradation. Unusual retention of genes and diverse rearrangements indicate that complex constraints affect the fate of the plastomes in heterotrophs. Although heterotrophs share a universal pattern of plastome degradation, the evolution of plastomes from autotrophs to heterotrophs may be species-specific or lineage-specific. In general, our study fills the gap of knowledge about the plastome of heterotrophs in Burmanniaceae, and it would be attractive to investigate plastomic evolution in the genus with more species and more populations within species to characterize the trajectories of plastomic evolution.

### Funding
This work was supported by the National Natural Science Foundation of China (Grants 31600185 and U1603231), and the Ministry of Science and Technology of China (Grant 2013FY111200). The funders had no role in study design, data collection and analysis, decision to publish, or preparation of the manuscript.

### Grant Disclosures
The following grant information was disclosed by the authors:
National Natural Science Foundation of China: 31600185 and U1603231.
Ministry of Science and Technology of China: 2013FY111200.

## Competing Interests

The authors declare that they have no competing interests.

## Author Contributions

- Xiaojuan Li conceived and designed the experiments, performed the experiments, analyzed the data, prepared figures and/or tables.
- Xin Qian performed the experiments, analyzed the data.
- Gang Yao performed the experiments, analyzed the data.
- Zhongtao Zhao conceived and designed the experiments, performed the experiments, analyzed the data, contributed reagents/materials/analysis tools, prepared figures and/or tables, authored or reviewed drafts of the paper, approved the final draft.
- Dianxiang Zhang conceived and designed the experiments, contributed reagents/materials/analysis tools, authored or reviewed drafts of the paper, approved the final draft.

## Field Study Permissions

The following information was supplied relating to field study approvals (i.e., approving body and any reference numbers):

The isolates of *Burmannia itoana*, the materials within our article (submission ID: 36139), were collected from the Longmen Park, Guangdong Province, China under local laws. No field permit was required as *B. itoana* is not a protected plant and the specimens were collected from a public park (Longmen Park) in small quantities that would not affect the ecosystem.

## Data Availability

Plastome of *Burmannia itoana* is available at GenBank: MK318822.

The voucher specimens (LXJLM07) are deposited in the IBSC (Herbarium of South China Botanical Garden).

## Supplemental Information

Supplemental information for this article can be found online at http://dx.doi.org/10.7717/peerj.7787#supplemental-information.

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
