# Peer review of "Plastome of mycoheterotrophic Burmannia itoana Mak. (Burmanniaceae) exhibits extensive degradation and distinct rearrangements"

_PeerJ, doi:10.7717/peerj.7787_

## Round 0.1 · original submission · Major Revisions

Please revise the manuscript as suggested by the reviewers. Please have a fluent English speaker correct the text to avoid language issues in the resubmitted manuscript.

[]

Reviewer 1 ·

Basic reporting

Please have a fluent English user examine the phrasing of the manuscript closely. The phrasing is could be misconstrued as speculative, e.g. "One nature of plastome is that its genes could be transferred or pruned." or "B. itoana plastome is not just downsizing..." or confusing "pseudogenized petG is retained"

Please review for misspellings e.g. "nonbioenirgitic", Burmannia is misspelled as "Burmania" twice, once in the abstract, once in line 41.

Line 50: Please either include all species representatives, or none. This is misleading.

Abstract: Inversion 2 (with accD) is touted as a result of a heterotrophic shift, which would be a very interesting result. This is nearly completely dropped in the discussion section, outside of lines 193 - 195.

Line 200: "Interestingly, the six rearrangements (Inv3/Inv4/Ins1/Ins2/Ins3/ins4) occur in IRs-SSC, IRs, IRs and IRs of B. itoana plastome, respectively" - I don't understand this sentence. Why are "IRs" repeated three times?

Experimental design

Line 54: I am not clear whether the DNA extracted was from the same specimen(s?) as those vouchered. It would also be useful if a better description could be given of the field site.

Line 61: please provide the sequences of these primers.

Validity of the findings

Line 80: Please specify your criteria for determining function vs. pseudogene

Additional comments

Lines 146 - 156: The identity of tRNA genes retained is interesting, and should be expanded.

Reviewer 2 ·

Basic reporting

This article meets all the basic standards of PeerJ.

Experimental design

The experiment of this study was well designed and the methods used are valid, described with sufficient details.

Validity of the findings

The analyses of this study are sound, and the results are informative. The sequencing of a plastome of heterotrophs in Burmanniaceae and comparative genomic analyses will fill the gap of knowedge about plastome evolution in this family.

Additional comments

The paper is publishable in PeerJ and I only have some minor comments:
1) The title is a bit long.
2) In line 81-82, how did you indentify these genes as pseudogenes? Could you provide some evidence such as frame-shift, stop codon mutations?
3) I suggest authors to consider to add a piturece of B. itoana into the figure, showing the heterotrophic lifestyle of this plant.

·

Basic reporting

The manuscript is quite concise and straightforward and I had no problem following its main message. I should note however that it could be greatly improved by the proofreading by the native speaker. Also, there are multiple typos, including in the plant names (e.g. Burmania instead of Burmannia).
Figures and tables are OK, with exception of Additional file 1 (see below).

Experimental design

The topic is interesting, the research question is well-defined and the results are valuable for the scientific community. The methods are adequate and desribed in sufficient details. However, as soon as one of the main findings of the manuscript is the observation of rearrangments in B. itoana plastome I suggest to perform more thorough analysis of the rearrangements, including the search for the repeats that could have trigger the rearrangements (see e.g. Weng et al. - 10.1093/molbev/mst257 and other studies of Geraniaceae plastomes which are highly rearranged and repeat-rich).

Validity of the findings

The data are robust, genome assembly is reliable. However, the annotation provided in the Additional file 1 is questionable - it looks like an uncorrected draft, with some notes on the results of the blast search and strange ORFs such as
gene complement(<319..>458)
/gene="atpI"
/note="blatX_hit atpI_Tacca_chantrieri_Copy.gb / position
619 - 728 / coverage 14.78% / match 11.16%"

ORF complement(611..1291)
/Genetic_code="Standard"
/Reading_frame=3

which do not go to the final annotation (as far as I can see from the Fig. 1). As soon as you submitted the annotated genome to NCBI I would suggest you to replace Additional file 1 by the GenBank flatfile.

Also, I have concerns about the introduction and discussion which are somewhat shallow. Several statements in the introduction and discussion are arguable.
e.g. lanes 36-37: "ndh genes become dispensable once plants have the ability of obtaining nutrients using heterotrophic way" - ndh genes are lacking or pseudogenized in many autotrophic plants (e.g. Peredo et al. 2013 10.1371/journal.pone.0068591). Knockout experiments on model plants also showed that they are dispensable under certain environmental conditions (10.1093/emboj/17.4.868).
lanes 198-200: "It was hypothesized that MORFFO (mobile open reading frames) elements are regularly associated with inversions and changes to the inverted repeats". First, the term MORFFO stands for "Mobile Open Reading Frames in Fern Organelles" and is thus by definition can be applied only to ferns. Second, and more important - MORFFO are unusual ORF, that do not have close similarity to typical plastid genes. As far as see from the results that you present (Fig. 1, in particular), there are no such elements in the B. itoana genome.

---

## Round 0.2 · Minor Revisions

Thank you for revising the manuscript based on the comments sent by the reviewers. Please revise the manuscript based on the additional comments provided by the Section Editor (below).

Comments by Section Editor:

“In reviewing the manuscript the Genbank accession MK318822 did not appear available; it was seen as a non-published supplement. GO annotations are ideal for breaking down terms into molecular, cellular, and biological classifications; it would be useful to have such information included as a supplemental table with the gene list assigned to appropriate GO: terms to associate this data with the plastome. Or such terms might easily be attached to a Table or attached to the gene lists as seem in Figure 3. Journal manuscripts are often scanned by text-mining software that locates and extracts core data elements, like gene function. Adding standard ontology terms, such as the Gene Ontology (GO, geneontology.org) or others from the OBO foundry (obofoundry.org) can enhance the recognition of your contribution and description. This will also make human curation of literature easier and more accurate. None of this was visible.”

---

## Round 0.3 · accepted · Accept

Thank you for making the additional changes that were requested.